# Perinatal Fat-Diets Increased Angiotensin II-Mediated Ca^2+^ through PKC-L-Type Calcium Channel Axis in Resistance Arteries via *Agtr1a-Prkcb* Gene Methylation

**DOI:** 10.3390/nu15010245

**Published:** 2023-01-03

**Authors:** Qiutong Zheng, Yun He, Lingjun Li, Can Rui, Na Li, Yumeng Zhang, Yang Ye, Ze Zhang, Xiaojun Yang, Jiaqi Tang, Zhice Xu

**Affiliations:** 1Institute for Fetology, First Hospital of Soochow University, Suzhou 215006, China; 2Nanjing Maternity and Child Health Care Hospital, Nanjing Medical University, Nanjing 210004, China; 3Maternal and Child Health Care Hospital of Wuxi, Jiangnan University, Wuxi 214122, China

**Keywords:** L-type-calcium-channel, DNA methylation, PKC, perinatal high-fat-diets

## Abstract

Perinatal malnutrition affects vascular functions, and calcium is important in vascular regulations. It is unknown whether and how perinatal maternal high-fat diets (MHF)-mediated vascular dysfunction occurs via the angiotensin-PKC-L-type-calcium-channels (LTCC) axis. This study determined angiotensin II (AII) roles in the PKC-LTCC axis in controlling calcium influx in the arteries of offspring after perinatal MHF. Mesenteric arteries (MA) and smooth muscle cells (SMCs) from 5-month-old offspring rats were studied using physiological, ion channel, molecular, and epigenetic analysis. Pressor responses to AII were significantly increased in the free-moving MHF offspring rats. In cell experiments, MA-SMC proliferation was enhanced, and associated with thicker vascular wall in the obese offspring. Imaging analysis showed increase of fluorescence Ca^2+^ intensity in the SMCs of the MHF group. Angiotensin II receptor (AT1R)-mediated PKC-LTCC axis in vasoconstrictions was altered by perinatal MHF via reduced DNA methylation at specific CpG sites of *Agtr1a* and *Prkcb* gene promoters at the transcription level. Accordingly, mRNA and protein expression of AT1R and PKCβ in the offspring MA were increased, contributing to enhanced Ca^2+^ currents and vascular tone. The results showed that DNA methylation resulted in perinatal MHF-induced vascular disorders via altered AT1-PKC-LTCC pathway in resistance arteries of the offspring, providing new insights into the pathogenesis and early prevention/treatments for hypertension in developmental origins.

## 1. Introduction

The perinatal environment is critical for developmental origins of health and diseases (DOHaD) [1]. Multiple antenatal stresses (hypoxia, malnutrition, and drug addiction), have been shown to cause a series of alterations in developing fetuses or neonates, leading to chronic adult diseases [2,3,4]. Over-nutrition such as maternal high-fat diets (MHF) during pregnancy and breast-feeding periods is common in modern societies. Several studies have indicated MHF increases blood pressure in the offspring [5]. However, the mechanisms underlying programmed vascular hyper-contractility have not been established in detail. Angiotensin II (AII) is an important regulator of cardiovascular functions, exerting diverse actions via its receptors AT1R and AT2R [6]. Previous studies demonstrated that AII increased blood pressure (BP) in MHF offspring [7], but the mechanism has not been elucidated. Our study determined the mechanisms and signaling pathways underlying AII-mediated vasoconstrictions in mesenteric arteries (MA) in the offspring rats.

In vascular regulation, the PKC-L-type calcium channel (LTCC) axis plays important roles in control of calcium influx in smooth muscle cells (SMCs) [8,9,10]. Our study paid attention to possible changes in that axis and calcium currents under perinatal MHF conditions. It is largely unknown whether and how LTCC or PKC may mediate AII-induced vascular changes by perinatal MHF. A previous report showed AII could increase blood pressure [11,12], but it is unknown whether and how AII might be involved in PKC-LTCC axis by perinatal MHF in peripheral vascular SMCs. The present study determined possible roles of the AT1R-PKC-LTCC axis in the resistance of arteries at both the tissue (vessel) and cell (SMCs) levels in the offspring exposed to perinatal MHF.

If AII-mediated vasoconstriction pathway was changed following exposure to perinatal MHF, the immediate question would be how the changes occurred. In recent studies on DOHaD, epigenetic mechanisms have been demonstrated repeatedly for important roles in pathophysiological processes [13]. DNA methylation has critical links to changes in gene expression [14]. However, it is unknown if perinatal MHF could affect DNA methylation in AT1R-PKC-LTCC axis and its functions. Therefore, the present study also determined DNA methylation in the promoters of AT1R and PKC genes and the expression of mRNA, as well as protein products of these genes.

## 2. Materials and Methods

### 2.1. Animals

Female rats (3-months old; Laboratory Animal Center, Shanghai, China) were mated with two male rats. The female rats showed vaginal mucus plugs on the morning after mating, were confirmed pregnant and the day was recorded as the first day of gestation. Pregnant Sprague-Dawley rats were divided into two groups randomly. The control group (CON) was provided with a standard rat diet (Catalog#: M01, Laboratory Animal Center, Shanghai, China), containing 22.1%protein, 52.0%mcarbohydrate, 5.3%fat, 1.0% sodium, 1.0% calcium, 0.9% phosphorus, 0.5% potassium, and 0.2% magnesium. The maternal high fat group (MHF) was fed the same diet except containing 60% kcal fat during the entire pregnancy and suckling period. All rats were housed in an environment with tap water and a 12-h light:12-h dark cycle. All male offspring were fed standard food after weaning and tested at 5 months old. They were weighed first. Rats were anaesthetized with isoflurane (5% for induction, 2% for maintenance) in oxygen (2 L/min for induction, 1 L/min for maintenance) and the adequacy of anesthesia was determined by the loss of a pedal withdrawal reflex as well as response to pinching the toe, tail, or ear of the animal. Rat heart and lung was isolated and weighed immediately and the second or third order of mesenteric arteries (MAs) were excised for vascular testing or histology analysis. All procedures conformed to the Guide for the Care and Use of Laboratory Animals (NIH Publication No. 85-23, 1996 and 2011 version) and approved by the Institutional Animal Care Committee of Soochow University (No: 2015291), and were carried out according to the guidelines from Directive 2010/63/EU of the European Parliament on the protection of animals used for scientific purposes.

### 2.2. Measurements of BP and Vascular Tension

Rats were anesthetized by intraperitoneal injection of 1% sodium pentobarbital at a dose of 30 mg/Kg and adequate anesthesia was determined by the loss of the pedal withdrawal reflex and other reactions from the animals in response to pinching the toe or tail. A polyethylene catheter (#BB31695-PE/1, 0.28 mm and 0.64 mm) was implanted in the femoral artery of adult offspring. Baseline BP was recorded for 60 min via an artery catheter one day after surgical recovery, then AII (100 ng/kg) was administrated via the catheter into femoral artery, BP was monitored/recorded for at least 30 min continuously using the Power Lab System (Power Lab 16/SP and Chart 5).

For vascular experiments, the second or third order of mesenteric arteries (MAs) (a separate cohort of animals) were excised and placed in physiological saline solution (mmol/L: NaCl, 120.9; KCL, 4.7; NaHCO_3_, 14.9; KH_2_PO_4_, 1.2; MgSO_4_·7H_2_O, 1.7; CaCl_2_·2H_2_O, 2.8; EDTA (ethylene diamine tetra acetic acid), 0.025; glucose, 5.0 and HEPES [2-[4-(2-hydroxyethyl) piperazin-1-yl] ethane sulfonic acid], 10.0 (95% O_2_, 5% CO_2_ for 40 min, pH 7.35–7.45), gassed continuously with 5% CO_2_ in O_2_). MA rings were threaded carefully onto two 40 μm diameter stainless steel wires and mounted in a 5 mL organ bath containing physiological saline solution. The system and equilibrating vessel rings were calibrated for at least 20 min, then KCL (60 mmol/L) was added at least twice to ensure that the vessels achieved optimum activity. Cumulative concentrations of AII (10^−11^~10^−5^ mol/L) were added, with Losartan (AT1R antagonist, 10^−5^ mol/L), PD123,319 (AT2R antagonist, 10^−5^ mol/L), GF109203X (PKC antagonist, 10^−5^ mol/L), 2-APB (IP3 antagonist, 10^−5^ mol/L), nifedipine (LTCC antagonist, 10^−5^ mol/L), Bayk8644 (LTCC agonist,10^−9^~10^−5^ mol/L) or with cumulative PDBU (PKC activator, 10^−11^~10^−5^ mol/L). All drugs were from Sigma-Aldrich (St. Louis, MO, USA).

### 2.3. Histological Experiment

The MA from 5-month-old offspring rats were fixed with 4% paraformaldehyde (Sigma-Aldrich (St. Louis, MO, USA), P6148) in PBS. Briefly, the vessels were fixed then embedded in paraffin and sectioned into slices of 1−3 μm on a microtome (Leica (Wetzlar, Germany), CM1850). Sections were deparaffinized at 60 °C for 30–60 min. Then, sections were treated in the following order: xylene (I), xylene (II), xylene (III), absolute alcohol, 95% and 75% ethanol for 1 min each. The sections were stained using hematoxylin solution (Bioworld Technology (St. Louis Park, MN, USA), BD5035), 1% hydrochloric acid alcohol, bluing reagent, 0.5% eosin (Solarbio, Beijing, China), G1100) 1 min, and 75, 95, absolute alcohol for 30 s. Sections were made transparent in xylene (I) and (II), and then sealed in neutral resin and dried in air. A microscope (Nikon, Tokyo, Japan), ECLIPSE, 80i) was used for observation and the thickness of the MA wall was measured with Image Manager software.

### 2.4. Electrophysiological Experiments

The third order of MA was separated and cut into small segments in Ca^2+^-free physiological fluid containing oxygen with 137 mmol/L NaCl, 5.6mmol/L KCL, 1.0 mmol/LMgCl_2_, 10.0 mmol/LHEPES, 0.44 mmol/LNaH_2_PO_4_, 0.42 mmol/LNa_2_HPO_4_, 4.2 mmol/LNaHCO_3_, and 10.0 mmol/L glucose; pH was adjusted to 7.4 with NaOH. Tissue segments were exposed to a series of enzymatic digestions (4 mg/ mL papain, 2 mg/mL bovine albumin, 1 mg/mL dithiothreitol) for 30 min at 37 °C, then the digested solutions were replaced by Ca^2+^-free PSS buffer three times and segments were blended gently using a pipette to release single muscle cells. All cells were stored at 4 °C. A conventional whole-cell patch clamp was used to record calcium currents. An AXON Multiclamp 700B with Clamped 10.1 was used for recording whole cell calcium currents. When cells were affixed to the glass bottom, Ca^2+^-free physiological fluids were replaced with fluids containing Ba^2+^ (i.e., 20 mmol/LBaCl_2_, 10 mmol/LEGTA, 10 mmol/L glucose, 1.0 MgCl_2_, 124 mmol/L choline-Cl) and adjusted to pH 7.4 with CSOH. The pipette (2–5 MΩ) solution consisted of 130 mmol/L cesium glutamate, 1.5 mmol/L MgCl_2_, 10 mmol/L HEPES, 10 mmol/L EGTA, 10 mmol/L glucose, 3 mmol/L Na_2_ATP, 0.1 mmol/L Na_2_GTP, and 0.5 mmol/L MgGTP; pH was adjusted to 7.4 with CSOH. BaCl_2_ was used to amplify calcium currents. Currents were elicited by a holding potential of −60 mV to test potentials in the range −60 to +50 mV with 10-mV increments. After recording the baseline of whole-cell Ca^2+^ currents, nifedipine (inhibitor of LTCC, 10^−5^ mol/L), GF109203X (PKC blocker, 10^−5^ mol/L), BayK8644 (activator of LTCC, 10^−5^ mol/L), AII (10^−6^ mol/L) or 5AZA (methylation inhibitor, 10^−5^ mol/L) + AII (10^−6^ mol/L) were added into the bath to test the calcium voltage. All the patch-clamp experiments were conducted using an Axon700B amplifier and Clampfit 10.1 software (Axon Instruments, Foster City, CA, USA). All calcium currents were digitized with a Digidata 1440A interface (Axon Instruments), with sampling at 10 kHz and filtration at 2 kHz. To measure the current density, current amplitudes were normalized by their membrane capacitance and expressed in pA/pF, calculated as the ratio of current amplitudes to capacitance.

### 2.5. Cellular Fluorescence Ca^2+^ Intensity

Isolated mesenteric myocytes were incubated with Fluo-3 AM (10^−6^ mol/L; Invitrogen, Carlsbad, CA, USA) in Ca^2+^-free physiological fluids for 30min. Then, myocytes were washed three times with 200 μl Ca^2+^-free physiological fluids. A total internal reflection fluorescence microscopy with electron-multiplying charge-coupled device imaging system was used to record single-myocyte images and changes of Ca^2+^ fluorescence in response to BayK8644 or AII. Nifedipine and GF109203X (10^−5^ mol/L) were used for pretreating myocytes for 15 min before application of AII (10^−6^ mol/L). All images were recorded and analyzed by Fiji software (Olympus, Tokyo, Japan). Ca^2+^ fluorescence intensity was calculated as F/F0 or ΔF(F-F0), where F0 is fluorescence intensity when the Ca^2+^ activity is stable, and F is the fluorescence intensity when reacting to the drugs.

### 2.6. Cell Culture and Tests

The offspring rats in both groups were anesthetized by intraperitoneal sodium pentobarbitone (2%, 150mg/kg), and then sacrificed by cervical dislocation. The MAs were collected quickly and dipped into pre-cooled sterilized PBS, gently swabbed to remove endothelial cells, cut into small pieces, then placed into 6 cm dishes with Dulbecco’s modified Eagle’s medium (DMEM, High Glucose supplement, HyClone (Hyde Park, UT, USA), SH30243.01), containing 15% FBS (Gibco, Waltham, MA, USA), 16140071), 100 mg/mL penicillin G, and streptomycin sulphate (Gibco, Waltham, MA, USA), 15140-122), and incubated in an incubator (37 °C, 5% CO_2_, Likang (Tainan City, Taiwan, China), HF90). After reaching 70–80% confluence, mesenteric smooth muscle cells (MA-SMCs) were passaged with 0.25% trypsin (Gibco, Waltham, MA, USA), 25200-056). Cells were used between three and five passages and were treated with Losartan (10^−5^ mol/L), GF109203X (10^−5^ mol/L), and PDBU (10^−5^ mol/L) for 48 h. Expressions of PKCβ were examined by Western blot analysis. Cell viability was assessed by the Kit-8 Cell Counting (CCK8, APE×BIO, K1018). Absorbance values were measured at OD450 nm with a Microplate Reader (Biosharp, Handan, China).

### 2.7. Quantitative Real-Time PCR and Western Blot

Total RNA was extracted from offspring MAs using Trizol reagent (Takara, Tokyo, Japan). Confirmation of the concentration and purity of extracted RNA was done by spectrophotometer (Bio-Rad, Hercules, CA, USA). First-strand complementary DNA (cDNA) was synthesized by the PrimeScript TM II 1st Strand cDNA synthesis kit (Takara) and the cDNA was diluted in nuclease-free water and stored at −20 °C until testing. Primer sequences are listed in Appendix A. The relative gene expression was calculated as 2^−ΔΔt^, using the reverse transcription products from the control as the calibrator to determine the relative expression level on the basis of the comparative ΔΔCt method. The protein abundance in MAs was assessed by Western blot analysis normalized to GAPDH. The primary antibodies were GAPDH (Proteintech, Rosemont, IL, USA; Cat#60004-1-1g, 1:10,000), AT1R (Proteintech, Cat#25343-1-AP, 1:1000), AT2R (Santa Cruz, Dallas, TX, USA, Cat#sc-9040, 1:200), Cacna1c (Proteintech, Rosemont, IL, USA, Cat#21774-1-AP, 1:1000), PKCβ (Proteintech, Rosemont, IL, USA, Cat#12919-1-AP, 1:1000). The secondary antibody included HRP-labeled Goat Anti-Rabbit IgG (H+L) (Beyotime, Haimen, China, A0208,1:5000) and HRP-labeled Goat AntiMouse IgG (H+L) (Beyotime, Haimen, China, A0216, 1:5000). An imaging system (Tanon-5200, Shanghai, China) was used to visualize bands and normalize the relative density of bands to GAPDH as a control using Alpha Ease FC software.

### 2.8. Genomic DNA Extraction and Methylation Sequencing

A whole genome DNA extraction kit (Vazyme, Nanjing, China) was used to obtain pure Genome DNA from offspring MA tissue (N = 5, and n = 10; each sample in N was mixed with MA tissue from two offspring for the methylation test, thus, n = 10). SDS-PAGE gel electrophoresis was used to check DNA quality. The DNA was diluted to a 20 ng/μL working concentration for sequencing. CpG islands located in the proximal *Agtr1a* and *Prkcb* gene promoter were selected according to the following criteria: (1) ≥60% ratio of observed/expected dinucleotide CpG; (2) ≥200 bp length; (3) ≥50% guanine-cytosine content. The primer sequences are listed in Appendix A. After polymerase chain reaction amplification, products were sequenced by Hiseq 2000 (Illumina, San Diego, CA, USA). Methylation level at each tested CpG site was calculated as the percentage of the methylated cytosines over the total tested cytosines. This experiment was performed by Tianhao Biotechnology Co, Ltd. (Shanghai, China), and the data were processed in a double-blind manner.

Data are available from the respective authors upon reasonable request. Information of major resources and detailed methods are provided in the Appendix A.

### 2.9. Statistical Analysis

All data are shown as mean ± S.E.M. Two-way analysis of variance (ANOVA) was used to analyze comparisons of concentration response curves and I–V curves, followed by the Bonferroni post-hoc test. The unpaired two-tailed Student’s *t*-test was adopted to analyze differences between groups. Curve fitting was performed with Graph Pad Prism 5 (GraphPad, San Diego, CA, USA) to analyze dose-dependent responses to treatments. Statistical significance was defined as *p* < 0.05.

## 3. Results

### 3.1. Weight, Vascular Wall Thickness and Tension, and Blood Pressure

Maternal high-fat diets increased offspring body weight, but heart and lung weight were not significantly changed. Vascular wall thickness of the MA in MHF offspring was significantly higher than that of the control (Figure 1A,B). In in vivo experiments withfree-moving, conscious offspring rats, systolic, diastolic and mean arterial pressure at baseline were significantly higher in the MHF offspring than in the control (Appendix A). The AII-induced pressor response was significantly higher in the MHF group than in the control (Figure 1C). In vitro vascular functional experiments on MAs showed that the maximum constriction to AII was 10% of the KCL-stimulated vascular constriction in the control, and about 25% of the MHF group, while KCL itself resulted in no differences in vessel tension. The pD2 (−log (50% effective concentration) in MHF offspring was higher, also (Figure 1D–F), as the curve shifted left. IP3 receptor inhibitor (2-APB) partially suppressed accumulative AII-increased vascular tension in the MA of both groups. There was no statistical difference in the suppressed levels between the two groups. Expression levels of mRNA in IP3 subtype1–3 in the adult offspring MA were the same in the two groups (Appendix A).

### 3.2. Angiotensin II-Mediated Contraction via AT1Ra and PKCβ in the Offspring MAs

AII-increased vascular tension in the MAs was inhibited completely by Losartan, but not PD123,319, in both the control and MHF offspring. Accumulative concentrations of the PKC agonist PDBU significantly increased MA tension, and the increased levels in the MHF were higher than those of the control. The PKC blocker GF109203X suppressed the AII-increased vascular tension in both the control and MHF, indicating that AII-mediated vascular constriction was mediated via the AT1R-PKC pathway (Figure 2A,C,D,F). Molecular analysis showed mRNA expression of *Agtr1a, Agtr2, Prkcb, not Agtr1b, Prkca, Prkce, Prkcd*, were significantly higher in the MHF-MAs than in the control. Western blot analysis showed that protein expression levels of AT1R, AT2R, and PKCβ were significantly higher in the MHF-MA (Figure 2B,E and Appendix A).

### 3.3. Membrane Ca^2+^ Channel Activation and Expression, Calcium Signaling

Electrophysiological recording showed that Ca^2+^ current density was significantly increased in the MHF group. The LTCC channel agonist BayK8644 increased Ca^2+^ currents in the offspring SMCs in both the control and MHF groups. The increased levels of MHF were higher than those of the control. The LTCC blocker nifedipine almost completely inhibited increased Ca^2+^ in the SMCs (Figure 3A). Calcium signal analysis showed that fluorescence changes induced by bayk8644 were higher in the SMCs of the MHF-MA than of the control (Figure 3B). Protein expression of Cav1.2 (a major unit of LTCC) was higher in the MHF (Figure 3C and Appendix A). Accumulative BayK8644-increased vessel tension was significantly higher in the MHF-MA. Nifedipine suppressed AII-increased vessel tone in the MAs of offspring in both groups (Figure 3D,E).

### 3.4. Angiotensin II Activated PKCβ in AT1R-PKCβ-LTCC Axis

In the control SMCs, PDBU did not affect cell viability, while GF109203X induced a decrease of cell viability. In the cultured SMCs of MHF-MA, PDBU increased cell viability while GF109203X inhibited cell viability (Figure 4A). Losartan significantly suppressed PKCβ protein level (Figure 4B,C) (Appendix A). Expressions of mRNA in RYR2 (a subtype of ryanodine receptors), Serca1 and Serca2 (subtypes of endoplasmic reticulum Ca^2+^ ATPase) in offspring MAs were significantly higher in the MHF-MA than in the control (Appendix A). Application of AII caused stronger fluorescence Ca^2+^ intensity in the MHF-MA cells. Pre-treatment with GF109203X or nifedipine did not result in statistical differences between the two groups (Figure 4D–F).

### 3.5. Calcium Current Changes in the AII-PKC-LTCC Axis

Patch clamp tests showed that AII-increased Ca^2+^ currents were inhibited by bothGF109203X and nifedipine in the MA-SMCs in both the control and MHF groups (Figure 5A–D). AII-mediated Ca^2+^ currents were higher in MHF group than that of the control (Figure 5E,F).

### 3.6. DNA Methylation of CpG Locus at Agtr1a and Prkcb Gene Promoters in the MAs

Sequence analysis identified two consecutive CpG islands that contain 23 CpG sites near the *Agtr1a* gene promoter (product location from 35957866 to 35958112, length 247; and from 35957640 to 35957884, length 245, chr17: NM_030985) (Figure 6A,B), and 32 CpG sites close to the *Prkcb* gene promoter (product location from 192233659 to 192233847, length 189; and from 192234218 to 192234387, length 170, chr1: NM_001172305) (Figure 6E,F). The bisulfate conversion efficiency of each sample was higher than 99%, indicating that the conversion was efficient and reliable (Figure 6D,H). Bisulfate sequencing showed that methylation levels in the MHF-MA were decreased in three and one specific CpG sites (7#,16#,18# in *Agtr1a* gene promoter; 30# in *Prkcb* gene promoter, Figure 6C,D,G,H). Appendix A show detailed sites and methylation levels of each CpG island.

### 3.7. The Methylation Inhibitor 5AZA Increased AII-Induced Ca^2+^ Currents

Angiotensin II-induced Ca^2+^ currents were increased by the methylation inhibitor 5AZA (Figure 7A–C) and statistical analysis showed significant differences (Figure 7D).

## 4. Discussion

We used rats as an MHF diet model, revealing the long-term impacts on vascular functions in the adult offspring. We demonstrated that perinatal MHF induced-hypertension in the offspring via the AT1-PKC-LTCC pathway with altered DNA methylation in mesenteric arteries in developmental origins. Specifically, DNA methylation played critical roles in perinatal MHF-increased AT1R expression and enhanced AT1R-activated PKCβ transcription by up-regulating AT1R and PKCβ expression in vascular SMCs, resulting in an over-activated AT1R-PKCβ-LTCC axis, leading to increased vascular tension and BP.

Previous studies showed that perinatal MHF caused an increase of birth weight [15,16]. Our study demonstrated the offspring body weight at 5 months old was significantly higher in the MHF group, although the heart and lung weight were similar between the control and MHF groups. However, litter number between the CON and MHF groups was similar, which indicated the increased birth weight in the newborn was not related to litter size (Appendix A). Notably, thickness of the MA wall in MHF offspring was greater than that in the control, providing new evidence that MHF not only influences body weight, but also has an effect on histological structures of mesenteric arteries, which may affect vascular functions and BP. Further in vivo testing on free-moving adult offspring demonstrated that both the baseline BP and AII-mediated BP responses were significantly increased in those rats with a thicker layer of MA wall. Our cell experiments showed a significant increase of cell viability in the SMCs from the MHF-MA group. Application of the PKC agonist PDBU further increased cell viability and the increased levels were significantly higher in the MHF offspring. It seems that enhanced viability of the SMCs in the MA might alter thickness of the vascular wall; PKC contributed to the increased cell viability in the SMCs.

Prior work documented the effectiveness of MHF in increasing BP in male adult offspring rats [17]. Concerning the underlying mechanisms, the elevated BP was attributed to elevated glucocorticoid levels due to abnormal pituitary negative feedback [18], or MHF affected the offspring brain to induce baroreflex dysfunction and sensitization of AII-induced hypertension [18]. While these findings are interesting, data on peripheral vascular dysfunction and mechanisms in perinatal MHF models is very limited. It is well known that the primary basis in development of hypertension is vascular dysfunction in resistance arteries such as mesenteric arteries [19]. In this regard, after confirming that perinatal MHF could cause an increase of AII-mediated pressor responses, we showed that the AT1R-PKCβ-LTCC signal pathway is involved in the vascular hyper-contractility in response to AII in the MHF offspring, adding new information to understand the pathological mechanisms of MHF-impaired vascular functions in relation to development.

As typical arterioles in the body, MAs determine the majority of pressor responses and influence BP. Angiotensin II is well known for its important vasoconstriction effects [20]. Therefore, the sensitivity of MAs to AII was determined in this study. Relative to KCL-induced constriction, AII–induced maximal constriction was about 10% in the control, and 25% in the MHF group, i.e., two-fold more than that of the control. Moreover, the increased maximal constriction and pD2 by AII in the MHF group was also higher, demonstrating that tension and sensitivity of the MA in response to AII was significantly altered by perinatal MHF. AII acts on its receptors in the cell membrane to produce vasoconstrictions via multiple pathways [21]. One typical route is via AT1R to the IP3 unit [22]. We determined this using the IP3 inhibitor 2-APB. 2-APB-suppressed AII-increased vascular responses did not show significantly differences between the two groups, suggesting that AII-increased hyper-contractility was independent of AT1R-IP3 in the MHF-MA group. We then turned to other possible pathways.

In vascular functional studies, AII and PDBU each produced significant tension in the MA of the control and MHF offspring, and the increased level was higher in the latter. Moreover, the AT1R antagonist losartan, and the PKC blocker GF109203X, could inhibit AII-increased MA tension, offering new evidence that perinatal MHF-enhanced vascular tone in the mesenteric arteries occurs via AII-PKC pathways in the offspring. It is well known that AT1R plays major roles in AII-mediated vascular tension, while effects of AT2R on vessel tension have been suggested to produce no, or weak, vasodilation [23]. In the present study, the AT2R antagonist PD123,319 had no significant effects on AII-produced maximum constriction.

Our results provide new evidence that vascular hyper-contractility in the mesenteric arteries of offspring was related to high expressions of AT1R and PKCβ. Analysis of mRNA and protein expression showed that AT1R and PKCβ were significantly increased in the MHF-MA. Although the expression of both AT1R and AT2R increased in the MHF group, apparently AT1R-mediated vasoconstriction was dominant. PKC is a major regulator of vascular functions, and increases PKC activities contributed to hypertension. Our study is the first to demonstrate that the AT1R-PKC pathway plays critical roles in perinatal MHF-increased vascular tension in the mesenteric arteries.

In cell experiments, electrophysiological analysis showed Ca^2+^ current density was significantly increased in the SMCs of MHF offspring compared to controls. The LTCC channel activator BayK8644-enhancd Ca^2+^ currents were greater in MHF SMCs than in the control. The increased currents could be suppressed by the LTCC blocker nifedipine, demonstrating that LTCC channel activities in the SMCs were altered in the MAs by perinatal MHF. Fluorescence Ca^2+^ imaging analysis also showed AII-induced and BayK8644-increased fluorescence Ca^2+^ intensity in the cells was stronger in the MHF group. Vascular test also showed that BayK8644-stimulated vessel tension in the MHF-MA group was greater than that in the control. By employing different inhibitors, this study demonstrated that when PKC or LTCC channels were blocked, AII-increased fluorescence Ca^2+^ signaling was suppressed in SMCs. These findings revealed that Ca^2+^ activities were significantly enhanced by perinatal MHF via the PKC-LTCC route in the mesenteric arteries.

For the first time, electrophysiological analysis, Ca^2+^ imaging, and vascular tests, showed that the LTCC channel activation-induced more Ca^2+^ in the SMCs of the MAs in MHF offspring. The expression of protein Cacna1c (a major LTCC unit) was significantly increased in the MHF-MA group compared to the control. Although previous studies have reported PKC-LTCC mechanisms in hypertension [24], our study is the first to show that the AII-AT1R-PKCβ-LTCC pathway is responsible for perinatal MHF-induced vascular hyper-activities, altered pressor responses as the increases risks of hypertension.

Studies to date have suggested that prenatal MHF is associated with increased BP, insulin mesenteric, dyslipidemia, obesity, and endothelial dysfunction in adult offspring [25,26]. Maternal high-fat diet-induced changes in the offspring were attributed to altered methylation of various genes, indicating that maternal nutrition has an extensive impact on the transcriptome of the offspring [27]. In the present study, novel epigenetic finding showed DNA methylation mediated changes in the AT1R-PKC-LTCC axis, which significantly impacted on AII-increased BP in the offspring. Sequence analysis identified 23 CpG sites within the *Agtr1a* gene promoter and 32 CpG sites within the *Prkcb* gene promoter. Bisulfite sequencing showed that methylation levels in the MHF-MA group were significantly decreased in three or one specific CpG sites at the *Agtr1a* or *Prkcb* gene promoter. Those specific CpG sites were very close to the transcription starting site (TSS) of the gene promoters, strongly indicating a link to increased *Agtr1a* and *Prkcb* gene expression, as well as AT1R and PKCβ protein levels in the MHF-MA group. A number of previous studies have also shown that if specific CpG sites at the transcription starting site of gene promoters were hypo-methylated, corresponding gene expression could be affected [28]. Therefore, our study found that hypo-methylation mediated *Agtr1a* and *Prkcb* gene expression may contribute to the abnormal axis and hyper-vasoconstriction in MHF offspring. In addition to MHF, excessive vasoconstriction of AII in other models such as hypoxia [29] and high-salt diet [30], have been associated with activated *Agtr1a* transcription. It is rational to speculate that the changes of AT1R and PKCβ caused by perinatal insults are of great significance to long-term changes of blood vessels in offspring. Notably, this is the first study to demonstrate that MHF in developmental origins may have an imprinting influence on the epigenome in the offspring’s mesenteric vessels.

## 5. Conclusions

This study provides ample evidence that AII is powerful in controlling Ca^2+^ currents via LTCC channels in vascular SMCs in resistance arteries, which can be altered by perinatal maternal foods and induce vascular dysfunction in the obese offspring. DNA methylation at specific CpG sites of *Agtr1a* or *Prkcb* gene promoters may be involved in alterations of the AT1-PKCβ-LTCC pathway, influencing Ca^2+^ currents and influx in the SMCs, leading to vascular dysfunction and increased blood pressure. The results also provide new insights into potential targets in early prevention and treatments of vascular diseases such as hypertension.

## Figures and Tables

**Figure 1 nutrients-15-00245-f001:**
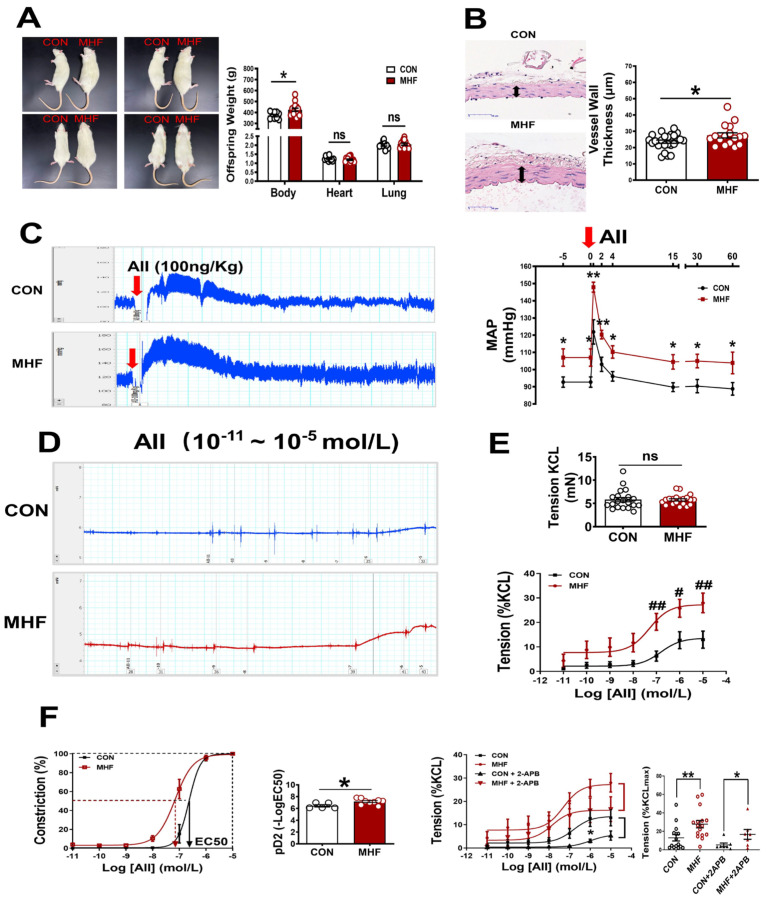
Maternal high-fat diet (MHF) affected offspring weight, blood vessel wall/tension and blood pressure. (**A**) Comparison of offspring body (N = 9 for CON, N = 15 for MHF), heart (N = 11 for CON, N = 15 for MHF), and lung weight (N = 10 for CON, N = 13 for MHF) between the control (CON) and the MHF group. (**B**) Thickness of mesenteric artery wall (N = 8 for CON, N = 6 for MHF), Arrows indicate thickness of the vessel wall. (**C**) Angiotensin II (AII)–mediated pressor responses. Arrow: time of AII injection (N = 5, per group). (**D**,**E**) Accumulative AII mediated vascular responses (N = 6 per group, n = 15 for CON, n = 16 for MHF). (**F**) EC50 and pD2 (n = 5 for CON, n = 9 for MHF), the effect of 2-APB on AII–mediated vasoconstrictions (N = 7, n = 7 per group). MAP, mean arterial pressure; N, number of litters; n, number of MA rings; MA, mesenteric arteries; 2-APB, IP3 inhibitor; ns, no significance; pD2, −log [50% effective concentration (EC50)]. Data were analyzed by Student’s *t*-test or 2-way ANOVA followed by Bonferroni Posttests. ^#,^* *p* < 0.05, ^##,^** *p* < 0.01.

**Figure 2 nutrients-15-00245-f002:**
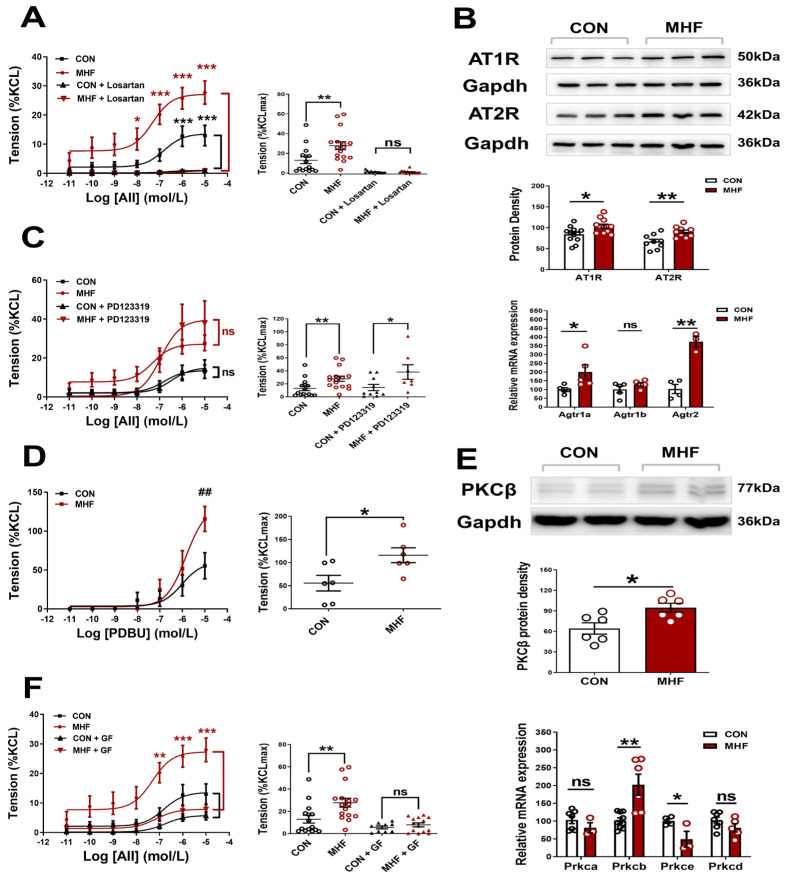
Angiotensin II (AII)-mediated vessel contraction via ATIR and PKCβ. (**A**,**C**), Cumulative AII–mediated vasoconstrictions in offspring mesenteric arteries (MA) after losartan (N = 6 per group, n = 12 for CON, n = 15 for MHF) or PD123319 (N = 6 per group, n = 10 for CON, n = 6 for MHF). (**B**) Protein (N = 6, n = 12 per group of AT1R, n = 9 of AT2R) and mRNA levels of *Agtr1a, Agtr1b* and *Agtr2* in offspring MA (N = 10, n = 1~3 MAs in each sample). (**D**,**F**), Cumulative AII–mediated vasoconstrictions in offspring MA after treatment with PDBU (N = 6, n = 6 per group) or GF109203X (N = 6, n = 9 for CON, n = 14 for MHF). (**E**) mRNA levels of PKC isoforms and protein levels of PKCβ in offspring Mas (N = 9, n = 1~3 MAs in each sample). GF, GF109203X; n, number of MA rings; N, number of litters; ns, no significance; Data were presented as mean ± SEM. Data were analyzed by Student’s-test or 2-way ANOVA followed by Bonferroni Posttests. * *p* < 0.05, ^##,^** *p* < 0.01, *** *p* < 0.001.

**Figure 3 nutrients-15-00245-f003:**
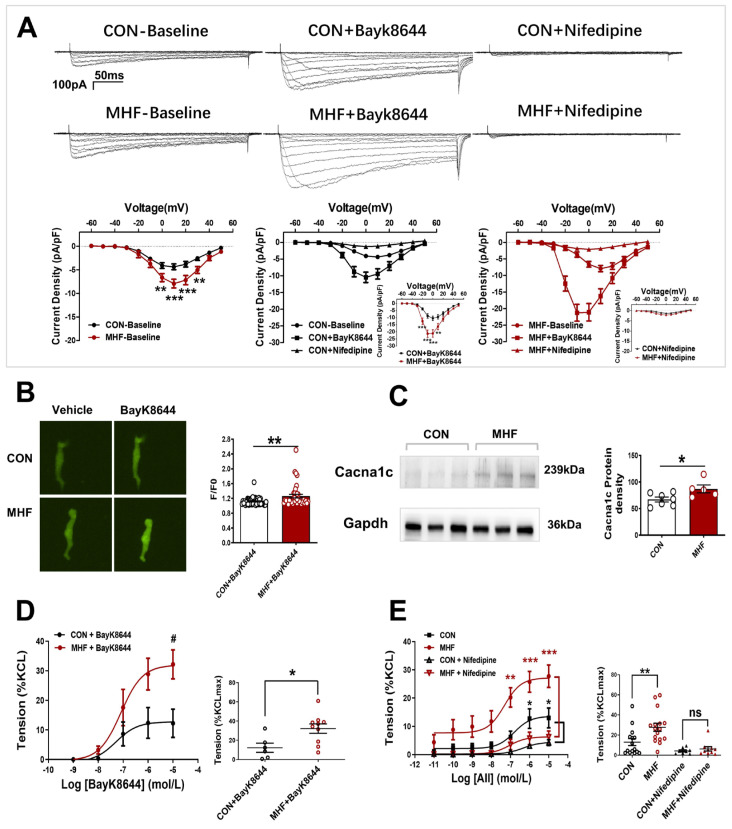
Altered L-type calcium channel activities and vasoconstrictions in offspring following perinatal fat diets. (**A**) Representative Ca^2+^ currents evoked (−60 to +50 mV, in 10-mV step)in the absence or presence of BayK8644 or nifedipine, and current density-voltage relationships of Ca^2+^ currents in the control and maternal high-fat diet (MHF) group (N = 6, n = 14 cells, each group). (**B**) Bayk8644–sparked Ca^2+^ was increased in mesenteric artery myocytes (N = 6, n = 46 for CON, n = 39 for MHF). (**C**) Expressions of Cacna1c protein (N = 7 for CON, n = 5 for MHF). (**D**,**E**) BayK8644-induced vasoconstrictions (N = 6, n = 6 for CON, n = 10 for MHF) and cumulative AII–mediated vasoconstrictions before and after treatment of nifedipine in offspring MA (N = 6, n = 8 for CON, n = 10 for MHF). N, number of litters; n, number of artery rings or myocytes. Data were analyzed by Student’s *t*-test or two-way ANOVA followed by Bonferroni post-tests. ^#,^* *p* < 0.05, ** *p* < 0.01, *** *p* < 0.001.

**Figure 4 nutrients-15-00245-f004:**
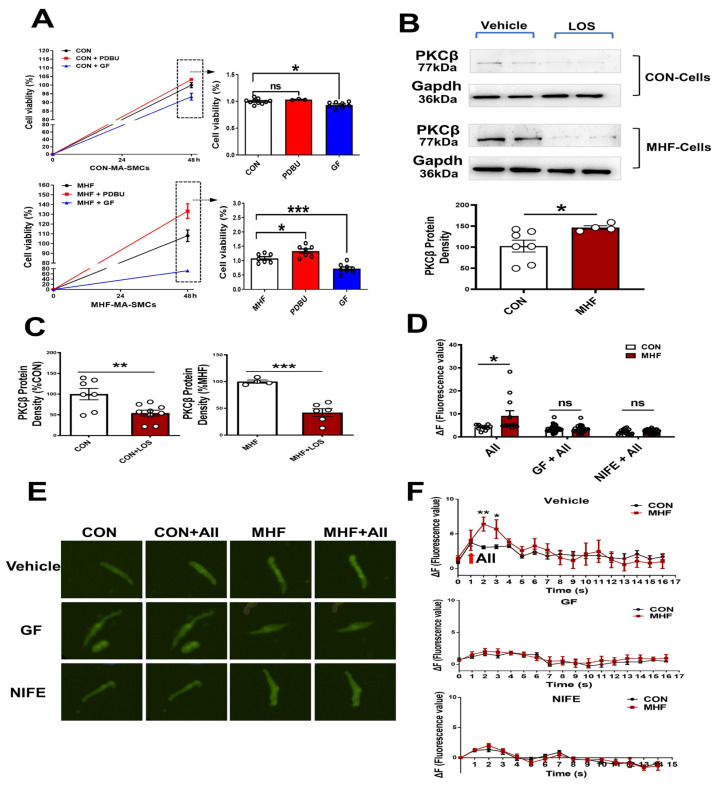
AngiotensinIIactivated PKCβ in the AT1R-PKCβ-LTCC axis. (**A**) Cell viability with or without PDBU or GF109203X, respectively (N = 6, n = 3~10). (**B**) Protein density of PKCβ in mesenteric artery cells. The cells were pre-treatment with vehicle (N = 7 for CON, N = 8 for MHF) or Losartan (N = 10 for CON, N = 6 for MHF). (**C**) Protein density of PKCβ. (**D**–**F**) Angiotensin II (AII)-induced an increase of Ca^2+^ in single isolated myocytes in the absence or presence of GF109203X or nifedipine. Representative fluorescence differences (F-F0) and traces of Ca^2+^ transients (N = 6, n = 11 or 12 of AII, n = 35 or 23 of GF+AII, n = 19 or 29 of NIFE+AII). GF, GF109203X; NIFE, nifedipine. N, number of litters; n, number of myocytes. Data were analyzed by Student’s *t*-test. * *p* < 0.05, ** *p* < 0.01, *** *p* < 0.001.

**Figure 5 nutrients-15-00245-f005:**
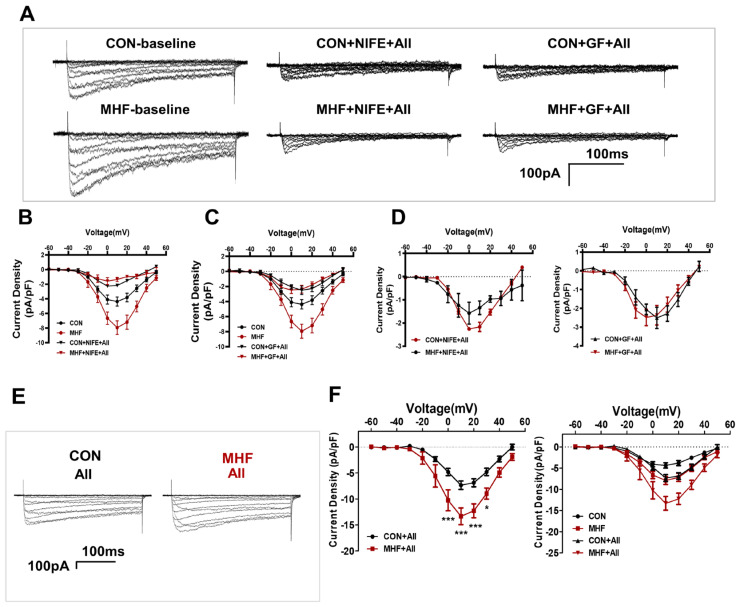
Calcium current changes in the AII-PKC-LTCC axis. (**A**) Representative images of AII-increased Ca^2+^ currents were inhibited by bothGF109203X and nifedipine in the MA-SMCs. (**B**–**D**) Statistics of AII-increased Ca^2+^ and currents inhibited by GF109203X and nifedipine. (**E**) AII-mediated Ca^2+^ currents in both groups. (**F**) statistics of AII-increased Ca^2+^ (N = 4, n = 3~4). N, number of litters; n, number of myocytes. Data were analyzed by two-way ANOVA followed by Bonferroni post-tests. * *p* < 0.05, *** *p* < 0.001.

**Figure 6 nutrients-15-00245-f006:**
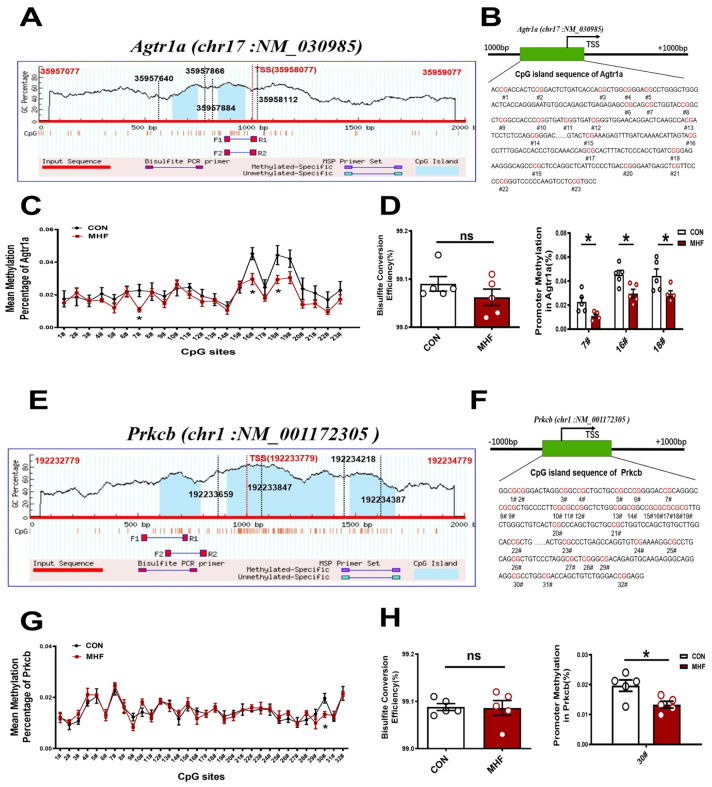
DNA methylation of the CpG locus at the *Agtr1a* and *Prkcb* gene promoter. (**A**,**B**) Bioinformatic analysis of CpG islands of *Agtr1a* from the upstream to downstream region. Sequence analysis identified two CpG islands that contain 23 CpG sites, with product location from 35957866 to 35958112, and 35957640 to 35957884 in the *Agtr1a* gene. Transcription starts site (TSS) were located at 35958077. (**C**) Methylation level at each site of Agtr1a. (**D**) Bisulfate conversion efficiency of each sample, and methylation level at #7, #16, #18. (**E**,**F**) Sequence analysis identified two CpG islands that contain 32 CpG sites close to the *Prkcb* gene promoter (product location from 192233847 to 192233659, and 192234387 to 192234218 in the *Prkcb* gene. TSS located at 192233779. (**G**) Methylation level at each site of *Prkcb* (N = 10 each group). (**H**) Bisulfate conversion efficiency of each sample and methylation level at #30. N, number of litters. Data were analyzed by Student’s *t*-test or two-way ANOVA followed by Bonferroni post-tests. * *p* < 0.05.

**Figure 7 nutrients-15-00245-f007:**
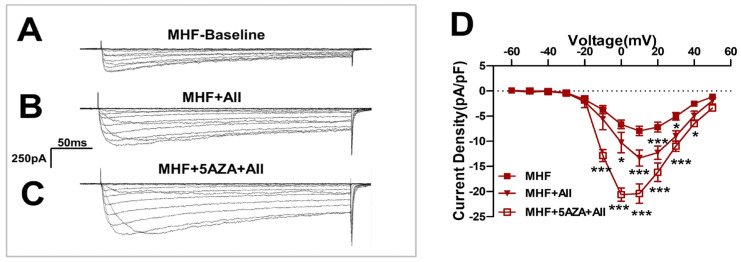
The methylation inhibitor 5AZA increased the AII-induced Ca^2+^ currents. (**A**–**C**) Angiotensin II(AII)-induced Ca^2+^ currents were increased by the methylation inhibitor 5AZA. (**D**) Statistics of patch clamp (N = 8, n = 5~14). N, number of litters. n, number of myocytes. Data were analyzed by Student’s *t*-test or two-way ANOVA followed by Bonferroni post-tests. * *p* < 0.05, *** *p* < 0.001.

## Data Availability

Original data in this study are available from the corresponding author according to reasonable request.

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
