# Peer review of "Perinatal Fat-Diets Increased Angiotensin II-Mediated Ca^2+^ through PKC-L-Type Calcium Channel Axis in Resistance Arteries via *Agtr1a-Prkcb* Gene Methylation"

_nutrients, 2023, doi:10.3390/nu15010245_

Round 1
Reviewer 1 Report
Review of manuscript ref. nutrients-2110248; Title: Perinatal fat-diets increased Angiotensin II mediated Ca2+ through PKC-LTCC axis in resistance arteries via Agtr1a-Prkcb methylation
Authors: Qiutong Zheng , Yun He , Lingjun Li , Can Rui , Na Li , Yumeng Zhang , Yang Ye , Ze Zhang , Xiaojun Yang , Jiaqi Tang , Zhice Xu.
General comment: the study reports the molecular mechanisms involved in the vascular dysfunction induced in 5 month-old offspring from malnourished mothers. The authors establish a model of perinatal maternal-high-fat-diets-mediated vascular damage and evaluate physiological parameters as well as calcium influx and molecular and epigenetic changes in mesenteric arteries and smooth muscle cells. Further, they investigate the molecular pathways potentially involved in the process, concluding that perinatal maternal high fat diet induced-hypertension in the offspring was via AT1-PKC-LTCC pathway with altered DNA methylation in mesenteric arteries. The hypothesis and objectives are sound and clearly exposed, methods seem adequate and results are interesting and with translational potential to humans. Some specific comments are detailed below:
Specific comments:
1) Although acronym for PKC is quite renowned, those for LCTT and Agtr1a-Prkcb are not widely known (at least this reviewer is not familiar with them) and should be spelled out or changed in title.
2) Figure 1B; the histological picture showing thickness of mesenteric artery wall seems to depict the contrary of the suggested result stated by the authors; thickness of control artery seems rather higher than that from MHF. The authors should indicate specific differences with arrows or other icons in order to support their conclusion.
3) Figure 3 and lines 270-273; although text in lines 270-273 states that calcium signal analysis showed that fluorescence changes induced by bayk8644 were statistically higher in the SMCs of the MHF-MA than that of the control, the truth is that differences in fluorescence intensity are not easily observed in figure 3B. Microscopic image should be representative of the quantified data from replicate assays; in any case, even if there is a significant difference, the physiological relevance seems scarce.
4) Figure 4B, same as above; western blot is not representative of densitometry data. Besides, if the purpose of the assay was to show the decrease induced by Losartan, densitometry data showing such decrease should be added to the figure.
5) In figures 1B, 2E, 6D,H, identification of data bars is repeated; colored icons on the upper right are not necessary.
6) English grammar should be revised in the whole manuscript.
Author Response
Response to Reviewer 1Comments
Point 1: General comment: the study reports the molecular mechanisms involved in the vascular dysfunction induced in 5 month-old offspring from malnourished mothers. The authors establish a model of perinatal maternal-high-fat-diets-mediated vascular damage and evaluate physiological parameters as well as calcium influx and molecular and epigenetic changes in mesenteric arteries and smooth muscle cells. Further, they investigate the molecular pathways potentially involved in the process, concluding that perinatal maternal high fat diet induced-hypertension in the offspring was via AT1-PKC-LTCC pathway with altered DNA methylation in mesenteric arteries. The hypothesis and objectives are sound and clearly exposed, methods seem adequate and results are interesting and with translational potential to humans. Some specific comments are detailed below:
Response 1: Thanks for the comments.
Point 2: Although acronym for PKC is quite renowned, those for LCTT and Agtr1a-Prkcb are not widely known (at least this reviewer is not familiar with them) and should be spelled out or changed in title.
Response 2: LTCC is for L-Type calcium channel, Agtr1a is the name of gene for angiotensin receptor subtype 1a, and Prkcb is also the name of gene for PKCβ. We changed LTCC in the title as suggested. For Agtr1a and Prkcb gene, they should be typed in italic type in both the title and text.
Point 3: Figure 1B; the histological picture showing thickness of mesenteric artery wall seems to depict the contrary of the suggested result stated by the authors; thickness of control artery seems rather higher than that from MHF. The authors should indicate specific differences with arrows or other icons in order to support their conclusion.
Response 3: A good question. We changed the representative figures, and marked the thickness of vascular wall with arrows as indicated by the reviewer.
Point 4: Figure 3 and lines 270-273; although text in lines 270-273 states that calcium signal analysis showed that fluorescence changes induced by bayk8644 were statistically higher in the SMCs of the MHF-MA than that of the control, the truth is that differences in fluorescence intensity are not easily observed in figure 3B. Microscopic image should be representative of the quantified data from replicate assays; in any case, even if there is a significant difference, the physiological relevance seems scarce.
Response 4: Excellent comment and we accept it. We changed the representative figures accordingly.
Point 5: Figure 4B, same as above; western blot is not representative of densitometry data. Besides, if the purpose of the assay was to show the decrease induced by Losartan, densitometry data showing such decrease should be added to the figure.
Response 5: We also changed the figure based on the comment. In addition, Figure 4C shows the calculated density for the Western blot data.
Point 6: In figures 1B, 2E, 6D, H, identification of data bars is repeated; colored icons on the upper right are not necessary.
Response 6: Very good point. We have delated the repeated identification in those figures.
Point 7: English grammar should be revised in the whole manuscript.
Response 7: Appreciate your comments and suggestions very much. We checked the language and grammar in the whole text. We also invited Prof. N. YU, Dept. of English, Foreign Language College of our University, to help our English editing for this manuscript. In case this revised and edited manuscript still requires further improvement in English, we would like to agree the suggestion of using the professional language editing service recommended by Nutrients later.

Reviewer 2 Report
See attached

Author Response
Response to Reviewer 2Comments
Point 1: In this study, the authors have examined the effect of a high fat diet during the perinatal period on hypertension in rats, with an emphasis on angiotensin II signaling as the mechanistic basis for the observed increase in blood pressure. Overall, the study design seems sound and the results clearly suggest that AII plays a direct role in this process.
Response 1: Thanks for the comments.
Point 2: The syntax and grammar need some work and at times it is difficult to follow the flow of the manuscript.
Response 2: Thanks for the suggestion, we did as suggested in the revision.
Point 3: Why were only males studied? Is there a reason to think the response would differ in females?
Response 3: An excellent question. It should be better if a study tested both male and female subjects, especially there may exist sex differences in the observed indexes. However, due to limitations, including limited budgets, the current work tested males first. In future investigations, it should be encouraged for study females too.
Point 4: There are sample sizes provided that refer to the number of litters, as well as the number of individual rats used within specific experiments. Initially, this was confusing, but it raises an important point. Due to the nature of the study, it is the individual litters that are the experimental subject. Sampling multiple individuals from the same litter but treating them as independent data points is a case of pseudo replication and inflates the power of the test used.
Response 4: Yes, the individual litters were the experimental subjects. Their fetuses or offspring were from different litters instead of the same litter. “N” were at least from 5-6 different litters in this study, which should be fine to statistically analysis. In addition, for patch clamp experiments using isolated single cells and vascular tests using vessel rings, many previous reports also used the similar “N” and “n” for number of litters and offspring or fetus.
Point 5: Were the same rats used to collect the vascular material that were used to collect the heart and lung tissue? It’s not clear how the experiments are described.
Response 5: Yes, we used the same rats to collect those vessels, heart and lung. The heart and lung were used only for measuring their weight. We added the information in the Methods based on the comment.
Point 6: Were the MA samples used for histology different from the ones used to collect the measures of vascular tension?
Response 6: Yes, after histological treatments, the MA samples cannot be used for other purposes. However, for adult offspring rats, there are sufficient MAs from individuals, some were used for histological test, the others can be used for testing vascular tension. We also added the information in the Methods.
Point 7: The text describing the electrophysiological experiments (section 2.4) is difficult to follow as written, especially for individuals that are not very familiar with these methodologies.
Response 7: A good suggestion. We rewrite the electrophysiological experiments based on the suggestion.
Point 8: For Figure 1A, it is not clear which rats are which.
Response 8: Agree. We marked which are which for Figure 1A in the revision.
Point 9: Figure 7E would be better as the first figure, maybe as a graphical abstract. Also, why was the effect of 5AZA not tested in control rats as well as the MHF rats?
Response 9: A nice suggestion, Figure 7E is removed from Figure 7 now, and acts as a graphical abstract. For DNA methylation experiments, 5AZA was used to test whether reversing effects would happen when this methylation inhibitor was introduced. Thus, the control group in this case was that MHF rats with hypomethylation responses, and the treatment group was that the same group but with additional pretreatment of 5AZA.
Point 10: The word “prove” is used in the discussion several times. These data don’t prove anything, they merely support the hypothesis.
Response 10: Thanks. Appreciate very much for correction of the misused word in English. We did corrections in the whole manuscript.
Point 11: The limitations paragraph in the discussion is not needed.
Response 11: Agree. The limitations paragraph was deleted in the revision.

Round 2
Reviewer 1 Report
Although authors responses are in Chinese and I could not understand them, by reading the revised version I check that the authors have conveniently addressed my comments and queries (English grammar revision is still needed) and, thus, I recommend to accept the revised version after extensive editing of English language.